# Development of the ECLIPSE model of meaningful outcome domains following lower limb amputation and prosthetic rehabilitation, through systematic review and best fit framework synthesis

**Chantel Ostler** [1,2]*, **Alex Dickinson**[3], **Cheryl Metcalf**[4], **Maggie Donovan-Hall**[2]

**1** Portsmouth Enablement Centre, Portsmouth Hospitals University NHS Trust, Portsmouth, United Kingdom,
**2** School of Health Sciences, University of Southampton, Southampton, United Kingdom, **3** School of Engineering, University of Southampton, Southampton, United Kingdom, **4** School of Healthcare Innovation and Enterprise, University of Southampton, Southampton, United Kingdom

* chantel.ostler2@porthosp.nhs.uk

## Abstract

### Background

Little is known about which outcome domains characterise meaningful recovery following prosthetic rehabilitation and should be measured. Our previous qualitative work developed a conceptual model of outcome domains which are meaningful to patients. This qualitative synthesis aims to develop that model by exploring views and experiences of recovery captured in the limb loss literature, and use these to produce a second iteration of the model describing outcome domains of importance following prosthetic rehabilitation from the patient's perspective.

### Methods

Systematic searches were conducted using CINAHL, Psychinfo and Web of Science from 2011 to early 2023. Studies with a qualitative design focusing on views and experiences of lower limb prosthetic users were eligible for inclusion. Quality was assessed using the CASP tool. 'Best Fit' framework synthesis was used to synthesise the evidence and develop the conceptual model.

### Results

40 studies were included, describing the experiences of 539 participants. Data supported the pre-existing conceptual model and led to development of four of the five domains. The newly named ECLIPSE model describes meaningful outcome domains as 1) Being able to participate in important activities and roles, 2) Participating in the *way* I want to, 3) My prosthesis works for me, 4) If I am in pain, I can manage it, and 5) I am able to accept my new normal. Studies came from 15 countries showing good coverage of high-income settings. Few participants from low-and-middle-income countries were included, it is unclear if the ECLIPSE model describes outcome domains of importance in these settings.

**Data Availability Statement:** All relevant data are within the manuscript and its Supporting Information files.

**Funding:** AD - funding from EPSRC/NIHR (grant ref EP/R014213/1). Funders played no role in the study CO - University of Southampton's Institute for Life Sciences. Funders played no role in the study. The funders had no role in study design, data collection and analysis, decision to publish, or preparation of the manuscript.

**Competing interests:** The authors have declared that no competing interests exist.

## Conclusions

This synthesis provides a rigorous foundation for understanding outcome domains of importance following lower limb prosthetic rehabilitation from the patient's perspective. The ECLIPSE model is an accessible representation of recovery which could direct rehabilitation programmes, as well as inform the evaluation of prosthetic care through the selection of outcome measures.

## Background

Outcome measurement is increasingly important in clinical practice, assisting clinicians to understand the impact of their interventions and the effectiveness of the services they provide [1]. The term outcome measurement can be better understood by breaking it down into i) the outcome domain being measured and ii) the measurement tool. An outcome domain can be defined as an element of health (i.e. pain, physical function, emotional wellbeing) that is changed by a particular intervention [2]. A measurement tool can be defined as a standardised instrument used in research and clinical practice to capture and evaluate change [3].

Despite its potential value, outcome measurement is still not routinely used in clinical practice [4]. Within prosthetic rehabilitation several clinical interest groups, such as the British Association of Physiotherapists in Limb Absence Rehabilitation (BACPAR) and the International Society of Prosthetics and Orthotics (ISPO), have attempted to increase health care professional engagement by publishing recommended outcome measures for use in clinical settings [5–8]. However, numerous outcome measures are included in the recommendations, with different measures proposed by different professional groups. The recommendations also include many outcome measures capturing the same outcome domain (i.e., mobility via measures such as the Six Minute Walk test, Timed Up and Go, or the Amputee Mobility Predictor). A recent narrative review highlighted the absence of outcome measure consensus in the field of prosthetic rehabilitation, and suggested it may be driven by a lack of understanding around which outcome domains characterise a meaningful recovery following prosthetic rehabilitation, and should therefore be measured [9].

Gaining consensus regarding outcome domains of importance is crucial to allow effective comparison of research findings and clinical data. Consensus is advocated for by organisations such as Core Outcome Measurement in Effectiveness Trials (COMET) [10], and the International Consortium of Health Outcome Measurement (ICHOM) [11], who recommend domain consensus in research and clinical settings, respectively. Both groups promote a multi-stakeholder approach, including patients, to ensure domains are relevant to those for whom health and rehabilitation interventions have the most impact. However, a recent review of patient participation in core outcome set development questioned how patient-centred the process is [12]. The review found health care professionals tended to dominate prioritisation exercises, and few studies employed qualitative methods that may give patients more opportunity to contribute in a meaningful way [12].

Within prosthetic rehabilitation several authors have begun to explore which outcome domains are important to people who use a prosthetic limb, using qualitative approaches. McDonald et al. and Shaffalitsky et al. [13,14] both explored outcome domains of importance following the prescription of a prosthesis, focusing on the impact of componentry rather than the wider, holistic impacts of prosthetic rehabilitation recommended by professional groups [5,15]. These authors identified domains of importance such as balance, independence and

adjustment, and interestingly highlighted *differences* in what patients and clinicians thought was most important [14]. Another small body of work attempted to develop an International Classification of Functioning (ICF) core set which could be used to inform which outcome domains to measure [16–19]. However, the authors identified several important concepts which could not be matched to the ICF and were therefore not included, such as socket comfort and feelings of acceptance following amputation. Moreover, the deductive approach recommended for ICF core set development may have diminished the voice of the patient.

This current study follows on from our large UK qualitative study [20], which began to address the knowledge gap regarding outcome domains of importance following prosthetic rehabilitation. The study included a heterogenous sample of 37 lower limb prosthetic users and identified five key outcome domains of importance from the patient's perspective, which were presented as a conceptual model to inform both outcome measure selection and rehabilitation priorities (Figure in S1 Appendix). The study included a wide range of views and experiences, but was limited by only involving individuals from the UK. Both convenience and purposive sampling were used to generate a diverse study population, however participants were identified by clinicians which may have led to a bias towards patients who had a positive experience of rehabilitation, or a more successful outcome.

Due to the limitations of a single qualitative study, further research is required to understand outcome domains of importance following prosthetic rehabilitation and continue developing the conceptual model considering the views and experiences of a larger population in different settings. Therefore, the aim of this article is twofold. Firstly, the study aims to use a systematic approach to search and synthesise published qualitative research, to explore outcome domains of importance following rehabilitation from the prosthetic user's perspective, as captured in the current evidence base. Second, the study extends the authors' empirical qualitative research described above [20] that underpinned the first stage of the conceptual model development, to generate a second iteration of the model informed by the wider experiences described in the limb loss literature.

## Materials and methods

### Research design

A systematic review of the literature and 'Best fit' framework synthesis were undertaken to address the research aims. The review was performed according to the PRISMA guidelines [21]. A comprehensive systematic approach was adopted to identify relevant publications, ensuring findings are based on a foundation of rigor and resonate with the prosthetic community which has been described as having a culture of quantitative enquiry [22]. 'Best fit' framework synthesis was used to analyse data and further develop the authors conceptual model of meaningful outcome domains in light of experiences described in the qualitative evidence base.

This approach was underpinned by a critical realist world view which looks to access the knowable world [23], in this case the perceptions of important outcome domains following lower limb amputation through the lens of prosthetic users. The conduct and reporting of this review adhere to the ENTREQ guidelines (Enhancing transparency in reporting the synthesis of qualitative research).[24]

### Search strategy

As recommended for aggregative approaches, such as 'best fit' framework synthesis, a systematic search strategy was undertaken to comprehensively identify all available studies and ensure that all possible data which may contribute to the synthesis were available [25–27].

**Table 1. Use of the SPIDER framework to define the search terms for the qualitative synthesis.**

| SPIDER tool | Search categories |
|---|---|
| Sample | Adults with lower limb loss |
| Phenomenon of Interest | Use of a prosthesis following lower limb amputation |
| Design | Any qualitative approach |
| Evaluation | views and experiences |
| Research type | Qualitative |

The SPIDER framework[28], adapted from the PICO framework for qualitative systematic reviews, was used to define the search terms (Table 1).

Following several scoping searches, the bibliographic databases CINAHL, Psychinfo and Web of Science were searched for relevant studies. These databases are recommended for use in qualitative syntheses as they have complete indexing for qualitative studies[25,29]. The search was limited to English language articles, published in peer reviewed journals. The Trip database was then searched to identify grey literature sources. Searches were limited to articles published in the last ten years between January 2011 and January 2023, to focus on current rehabilitation services and advances in prosthetic technology, and any shifts in societal acceptance of disability. The search strategy is described in Table 2.

## Screening process

Two reviewers (CO and AD) undertook title and abstract screening using Rayyan, a web application for systematic reviews (Rayyan Systems Inc.). Following the removal of duplicates, CO

**Table 2. Search strategy used for qualitative systematic review.**

| Database | Syntax |
|---|---|
| CINAHL | ((Amput* OR prosthe* OR "limb loss" OR "artificial limb*") OR (MH "Amputation" OR MH "Above-Knee Amputation" OR MH "Amputation Stumps" OR MH "Below-Knee Amputation" OR MH "Disarticulation" OR MH "Hemipelvectomy") OR (MH "Prosthesis Design" OR MH "Limb Prosthesis")) **AND** (("lower limb*" OR leg*) OR (MH "Lower Extremity" OR MH "Ankle" OR MH "Hip" OR MH "Knee" OR MH "Leg" OR MH "Thigh") OR (MH "Leg")) **AND** ((Qualitative OR experience* OR interview* OR "grounded theor*" OR phenomenolog* OR "focus group*" OR narrative OR "thematic analysis" OR "Action research" OR ethnograph*) OR (MH "Qualitative Studies" OR MH "Action Research" OR MH "Ethnographic Research" OR MH "Ethnological Research" OR MH "Ethnonursing Research" OR MH "Grounded Theory" OR MH "Naturalistic Inquiry" OR MH "Phenomenological Research") OR (MH "Life Experiences" OR MH "Work Experiences") OR (MH "Semi-Structured Interview" OR MH "Interview Guides" OR MH "Unstructured Interview" OR MH "Unstructured Interview Guides" OR MH "Structured Interview" OR MH "Structured Interview Guides" OR MH "Interviews") OR (MH "Focus groups") OR (MH "Narrative medicine") OR (MH "Thematic analysis")) |
| Psychinfo | ((Amput* OR prosthe* OR "limb loss" OR "artificial limb*") OR (DE "Amputation" OR DE "Prostheses" OR DE "Phantom Limbs")) **AND** (("lower limb*" OR leg*) OR DE "Thigh" OR DE "Ankle" OR DE "Knee")) **AND** ((Qualitative OR experience* OR interview* OR "grounded theor*" OR phenomenolog* OR "focus group*" OR narrative OR "thematic analysis" OR "Action research" OR ethnograph*) OR (DE "Focus Group Interview" OR DE "Focus Group" OR DE "Grounded Theory" OR DE "Interpretative Phenomenological Analysis" OR DE "Narrative Analysis" OR DE "Semi-Structured Interview" OR DE "Thematic Analysis" OR DE "Phenomenology") OR (DE "Experiences (Events)" OR DE "Life Changes") OR (DE "Action Research") OR (DE "Ethnography")) |
| Web of Science | (Amput* OR prosthe* OR "limb loss" OR "artificial limb*") **AND** ("lower limb*" OR leg*) **AND** (Qualitative OR experience* OR interview* OR "grounded theor*" OR phenomenolog* OR "focus group*" OR narrative OR "thematic analysis" OR "Action research" OR ethnograph*) |
| Trip database (Grey literature) | Amputation **AND** Prosthesis **AND** qualitative |

**Table 3. Inclusion and exclusion criteria used for screening of articles.**

| **Inclusion Criteria** |
|---|
| Adult populations 18yrs and older |
| Included participants with a major lower limb amputation (At level of ankle and above) |
| Included prosthetic limb users |
| Use of qualitative study design (i.e. interviews, focus groups, grounded theory etc.) |
| Studies exploring views and experiences of life with a prosthetic limb |
| Presenting first person accounts |
| **Exclusion Criteria** |
| Included participants with upper limb or minor lower limb amputations (i.e. toes or partial foot) or studies which combined these populations with major lower limb amputations |
| Included those not using a prosthetic limb or studies which combine these populations with limb wearers |
| Studies only exploring prosthetic service provision |

screened all articles with AD screening a random sample of 13% of abstracts. Agreement between reviewers was 99.6% with a single paper requiring discussion before it was excluded. CO then undertook full text screening using the inclusion and exclusion criteria (Table 3). Studies including mixed populations, i.e., prosthetic, and non-prosthetic users, were only included if data specific to the population of interest was presented independently in the analysis to ensure the outcome domains of importance were relevant to lower limb prosthetic users. Undecided papers were reviewed by AD and MDH and agreed upon following discussion.

## Critical appraisal

Critical appraisal within a qualitative synthesis is controversial [29]. Researchers dispute whether or not to undertake it, *how* to do it, whether to exclude studies as a result of it, and finally how to integrate critical appraisal findings into the main body of the synthesis [25,30]. Despite these questions there is a growing trend towards including critical appraisal within a qualitative synthesis, and it is recommended as part of the 'best fit' framework synthesis approach [26,27].

The critical appraisal process was used to give context to the findings presented in the synthesis, and comment on the quality of the overall sample [26,27]. The CASP tool [31] was used to undertake critical appraisal. Initially 10% of the papers were appraised by two reviewers (CO and MDH) to set quality expectations within each CASP question and compare and agree on the appraisal approach. CO then continued to appraise the remaining papers seeking advice and agreement from MDH where required. To summarise the findings, each quality appraisal response from the CASP tool was allocated a score from 1–3 (1 = yes, 2 = can't tell and 3 = no). No studies were excluded due to perceived poor quality, to ensure all possible outcomes of importance were considered at this stage, and instead they were ranked in terms of quality.

## Data extraction

Data extraction was undertaken by CO in two stages. Firstly, study-related data were extracted including the aim, design, sample size, recruitment setting, data collection method and geographical location, as well as details about the included population such as time since amputation, cause of amputation, sex, level of amputation and age range. Data were extracted to describe the studies and the characteristics of the study samples.

The second phase of data extraction addressed the qualitative findings of the included studies. Data were considered as that which were presented in the results or findings sections of the papers, and included both verbatim quotations and interpretations made by the study

authors which were clearly supported by the study's data [30]. Data were imported into NVIVO software (QSR International, Melbourne, Australia) for analysis.

## Stages of analysis

**Stage 1. Framework development.** 'Best fit' framework synthesis [26] uses an 'a priori' framework based on an existing conceptual model to synthesise study data and examine and develop new iterations of the model based on findings from the wider literature. An initial conceptual model of outcome domains of importance was developed by these authors using a primary qualitative approach to explore the lived experience of prosthetic users and is published elsewhere[20]. This work involved interviews and focus groups with thirty-seven lower limb prosthetic users from four English prosthetic centres. Data were analysed using reflexive thematic analysis to develop five themes, with ten associated subthemes, which describe outcome domains of importance from the patient's perspective. The five themes were visualised into an initial conceptual model (Figure in S1 Appendix). This first stage model acted as the pre-existing conceptual model underpinning the 'a priori' framework, and for clarity will now be referred to as the pre-existing model.

An *'a priori'* framework was developed (Table 4) by deconstructing the pre-existing model into its comprising themes and subthemes. This created an in-depth framework grounded in the findings from the authors previous qualitative study [20]. The themes, referred to in the framework as domains were described using first person statements to ensure that the voice of

**Table 4. Domains from the pre-existing conceptual model, including detailed definitions, which make up the 'a priori' coding framework.**

| Framework domain | Definition |
|---|---|
| **Domain 1**—I am able to participate in my important activities | |
| 1.1 Walking again | *Walking is the first step in the recovery process and is important in feeling normal again* |
| 1.2 Important activities at home | *Being able to do household tasks again, in a standing position, and get out of the house, even if only into the garden* |
| 1.3 Important activities in my community | *Being able to undertake whatever activities are important to me, and having the mobility skills i.e. on uneven ground and slopes, to be able to do so* |
| **Domain 2**—I can participate in my important activities in the *way* I want to | |
| 2.1 Doing my activities independently | *Being able to do important activities independently without having to rely on anyone else* |
| 2.2 Doing my activities easily | *Mastering my important activities so I don't have to think about what I'm doing and I feel confident doing them.* |
| 2.3 Doing my activities without falling over | *I can do my important activities without falling over, or fear that I will fall, and I can get up on my own if I do fall.* |
| 2.4 Doing my activities with as little equipment as possible | *I only use equipment that I really need to allow me to do my important activities. Less equipment makes me feel more normal* |
| **Domain 3**—My prosthesis is comfortable and easy to use | *My prosthesis is comfortable to wear for as long as I need, and for the different activities I want to do. It does not damage my skin or make me too sweaty. My prosthesis is easy to get on and off and not too burdensome to use throughout the day as the fit changes* |
| **Domain 4**—If I have pain, I am able to manage it | *If I have pain I can manage it in a way that enables me to accept and live with it.* |
| **Domain 5**—I am able to accept my new normal | |
| 5.1 Chasing normality | *I feel I am back to normal and the person I was before the amputation* |
| 5.2 Adjusting to limb loss | *Adjusting is hard but my family and I have adjusted to the amputation and are able to accept what I can do now and how I now look* |
| 5.3 Sense of achievement | *I have achieved my goals and feel proud of myself. I will continue to set goals in the future.* |

the prosthetic user was not lost during the synthesis process. Each framework domain was also accompanied by an in-depth description to aid consistency of coding [26,27].

**Stage 2: Analysis.** Data describing the included studies and their samples were analysed using descriptive statistics to give context about the qualitative approaches taken and the overall review population.

Data synthesis from the 'findings' sections of the included articles was undertaken in two steps. Step one involved open line by line coding of the data, codes were then mapped onto the domains and subthemes described in the 'a priori' framework (Table 4).

Codes that did not fit easily into the framework were collated separately in NVIVO and analysed in a second step, independent of the framework synthesis, using thematic analysis as described by Braun and Clarke [32–34]. This dual approach using inductive thematic analysis in addition to the more deductive framework synthesis (Table 5) allowed previously unidentified concepts related to outcome domains of importance to arise from the data.

**Stage 3: Conceptual model development.** The findings from the framework synthesis were reviewed by the research team to understand where the review data supported pre-existing domains and where they did not. Newly identified themes were reviewed against the preexisting conceptual model and through discussion and reflection, were added or used to refine the model until consensus was reached on a second iteration.

## Results

### Summary of included studies

Searches identified 2709 records, which were filtered down to 101 potentially relevant articles following removal of duplicates and screening of titles and abstracts. Thirty-nine of

**Table 5. Description of 'Best fit' framework synthesis and accompanying thematic analysis.**

| Phase | Description of process |
|---|---|
| (1) Familiarisation with the data | The results sections of the included studies were read and reread to increase familiarity with the data (CO). |
| (2) Coding | Open, line by line coding of the data was performed separately by the lead author (CO). Extracts of text were coded in as many ways as needed. A reflective journal was completed throughout the analysis process to encourage awareness of the researcher's own views and assumptions (CO). |
| (3) Coding into the framework | Codes were reviewed and mapped onto the domains and subthemes described in the 'a priori' framework by two researchers (CO and MDH). Data which did not map easily into the framework were collated separately. |
| (4) Reviewing left over codes | For codes not easily represented by domains set out in the framework a thematic analysis was undertaken. Left over codes and coded data were examined (CO and MDH), similarities and overlap were identified between codes and potential patterns relevant to the research question were created (CO and MDH) |
| (5) Generating and developing new themes | A visual map of initial themes not represented in the framework was created and compared (CO and MDH). All results sections were re-read and the fit of initial themes reviewed in relation to the full data set, coded data and the framework (CO). This process was then repeated by members of the research team (MDH). |
| (6) Refining, defining and naming new themes | The full set of concepts from both the framework and the additional thematic analysis were then reviewed and refined. Themes were collapsed or expanded in order to present coherent patterns within the data (CO). The research team reviewed newly developed concepts and subthemes to ensure they captured important new meaning in relation to the research question, and to assist reflection on researcher assumptions (CO, AD, CM, MDH). A person-centred approach was taken to naming new domains and subthemes in order to capture the voice of participants (CO). |

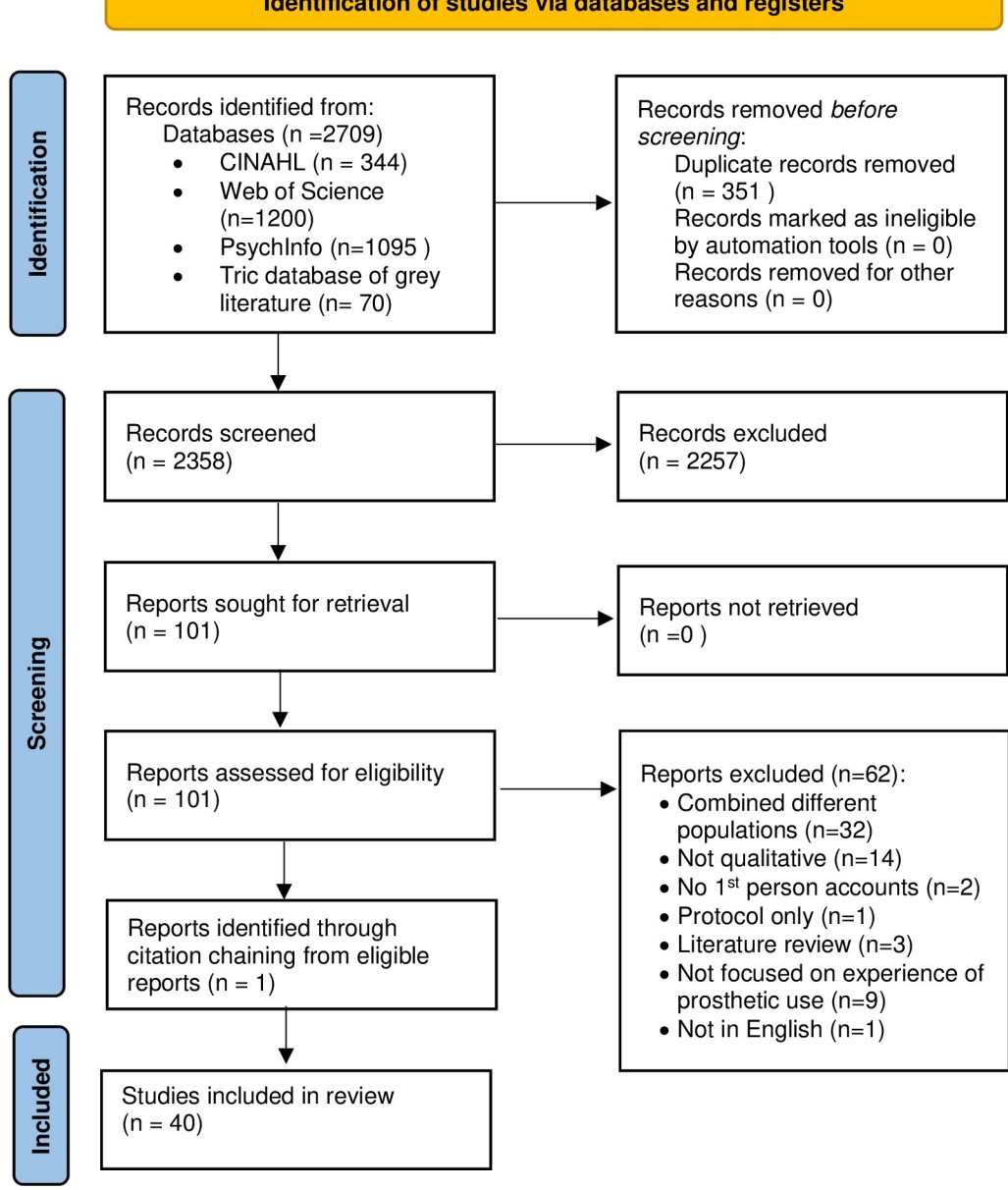

**Fig 1. PRISMA diagram describing the process of identifying, screening and selecting articles for inclusion in the qualitative synthesis.**

these studies met the inclusion criteria, with an additional study identified via citation chaining (Fig 1).

The studies identified explored the experiences of 539 participants, 193 of whom were female (35.8%). Demographic data regarding cause of amputation were available from 31 studies representing 444 participants (82.3%). Across all studies, the causes of amputation were trauma (n = 206, 46.4%), diabetic dysvascularity (n = 130, 29.3%), cancer (n = 44, 10%), infection (n = 37, 8.3%) and congenital aetiologies (n = 6, 1.4%). Demographic data describing level of amputation were available from 34 studies (n = 499, 92.6%). The levels of amputation were transtibial level (n = 286, 57.3%), transfemoral (n = 128, 25.7%), ankle (n = 13, 2.6%), knee

(n = 22, 4.4%) and hip (n = 6, 1.2%) disarticulation amputations. Forty three participants experienced bilateral limb loss (8.6%). The age of participants ranged between 18–81 years. The study aims and sample characteristics are described in Table 6.

The studies were undertaken in 15 different countries (Fig 2), with 486 (90.2%) participants living in high-income countries, according to the World Bank definition [74]. Eleven (2%) participants lived in upper middle-income countries, 36 (5.7%) in lower middle-income countries and 6 were not stated (2.1%). No participants were included from low-income countries.

## Methodological quality of included studies

The quality of papers included varied considerably. Overall, there was a little consideration of the relationship between the researcher and the participants, which was only adequately discussed in 12 of the 40 studies. Critical examination of the potential influence the researcher may have is important to provide insight into how their assumptions may have impacted or introduced bias to the results [75].

The other notable quality concern was recruitment of participants. Only 23 of the studies adequately described why participants selected for the study were appropriate to answer the research question. Many studies used convenience sampling approaches which may have led to samples with little variation, which do not represent the characteristics of target population [35,43,49,51,53–55,59,66,68,70,73].

CASP scores (out of 27, higher indicating poorer quality) ranged from 9–22. It is important to note that the findings described in this paper are supported by articles scoring across this range. The results of the critical appraisal process are summarised in (Table 7).

## Best fit framework synthesis

The 'best fit' framework synthesis illustrated that the experiences discussed within the included papers, undertaken in a variety of contexts, fit well into the pre-existing conceptual model. All of the model's domains were supported by the qualitative data (Table 8).

## Additional thematic analysis

Although additional data were identified which did not fit easily into the framework, following thematic analysis it became clear that they expanded the existing outcome domains of importance, rather than describing new ones. The next sections illustrate how each domain has been re-specified or developed, and provide additional context from the synthesis with relevant quotations. Domain changes are identified in bold underlined text within the following tables.

**Domain 1—I am able to participate in my important activities and roles.** The first domain 'I am able to participate in my important activities', set out in Table 9, was expanded to include an additional subtheme describing the importance of returning to valued roles. Roles were not included in the pre-existing framework which focused on participating in important activities.

*Subtheme 1.4—Fulfilment of roles*. The subtheme of role fulfilment is linked to 'participation in important activities', and was discussed in 13 papers [13,35,40,41,43,44,46,49,62,66,67,71,72]. Fulfilment of a role such as a parent, spouse or valued member of the community, was described as an important outcome following lower limb amputation. Role fulfilment was described alongside valued activities, but additional meaning was apparent when participation in the activity enabled previously valued self-identities, for example, holding a partner's hand when out for a walk, or being able to go to the park and play with their children.

**Table 6. Summary of study aim and sample characteristics from papers included in the qualitative synthesis.**

| Study | Aim | Location | Sample size (n) | Sample characteristics |
|---|---|---|---|---|
| Abouammoh et al. (2021) [35] | Explore the adjustment experiences of amputees in Saudi Arabia and their needs before and after amputation | Saudi Arabia | 8 | 5 females. Level: 1 symes, 3 TTA, 9 TFA, 6 bilat. Age range 26–71 yrs. Time since amp 4–15 yrs |
| Batten et al. (2020) [36] | Investigate barriers and enablers to community walking among people with lower limb amputation who have returned to live in a community setting | Australia | 14 | 5 females. Cause: 1 trauma, 9 diabetic dysvascular, 2 infection, 1 cancer, 1 other. Level: 13 TTA, 1 TFA, 2 bilat. Median age 58 yrs. Age range 49–62 yrs. Time since amp 4–24 months |
| Bragaru et al. (2013) [37] | Identify personal barriers and facilitators that influence participation in sports of individuals with LLA | Netherlands | 26 | 7 females. Cause: 7 trauma, 15 diabetic dysvascular, 4 tumour. Level: 1 symes, 9 TTA, 7 KDA, 7 TFA, 2 HAD, 2 bilat. Age range 21–77 yrs. Time since amp 2–35 years. |
| Camacho et al. (2021) [38] | Explore the lived experience of support group participants who are survivors of LLA living with PLP and understand the adaptation process postoperatively | USA | 10 | 6 females. Cause: 2 trauma, 5 diabetic dysvascular, 1 infection, 1 tumour, 1 congenital. Level: 4 TTA, 5 TFA, 1 HAD, 1 bilat. Age range 22–70 yrs. Time since amp 1–53 yrs. |
| Christensen et al. (2018) [39] | Increase understanding of the military identity influence on the organization of rehabilitation and investigate factors of importance for successful rehabilitation services | Denmark | 6 | All male. Level: 5 TTA, 1 TFA. Mean age 32 yrs. Age range 25–46 yrs. Time since amp 2–17 yrs |
| Crawford et al. (2016) [40] | Investigate barriers and facilitators to Physical Activity participation for men with transtibial osteomyoplastic amputation | USA | 9 | All male. Level: all TTA. Age range 31–35 yrs. Time since amp 2–33 yrs |
| Day et al. (2019) [41] | Explore the everyday experiences of people with an amputation using a good day/bad day approach | UK | 22 | 14 females. Cause: 9 Trauma, 3 diabetic dysvascular, 5 infection, 4 tumour, 1 congenital. Level: 18 TTA, 4 TFA, 4 bilat. Mean age 42 yrs. Age range 23–60. Time since amp 1–27 yrs |
| Devan et al. (2015) [42] | Explore the perceptions of adults with lower limb amputation and LBP as to the factors contributing to and affecting their LBP. | New Zealand | 11 | 3 female. Level: 8 trauma, 1 tumour, 1 congenital, 1 other. Level: 8 TTA, 3 TFA. Age range 18–70 yrs. Time since amp 3–54 years |
| Ennion and Manig (2019) [43] | Explore the experiences of current lower limb prosthetic users in relation prosthetic service delivery and the value of their prosthesis in a rural setting | South Africa | 9 | 1 female. Cause: 8 trauma, 1 infection. Level: 3 TTA, 6 TFA. Mean age 44 yrs. Age range 33–64 yrs. Time since amp 3–34 yrs |
| Hafner et al. (2016) [44] | Explore how prosthetic limb users conceptualize mobility with a prosthetic limb, construct a conceptual model of prosthetic mobility meaningful to people with lower limb loss, confirm key definitions, and inform development of items for the Prosthetic Limb Users Survey of Mobility (PLUS-M) | USA | 37 | 11 females. Cause: 25 trauma, 3 diabetic dysvascular, 11 infection, 2 tumour, 3 other. Level: 25 TTA, 1 KDA, 11 TFA, 1 HDA, 9 bilat. Mean age 50.4 yrs. Age range 22–71 yrs. Time since amp 0.5–60 yrs |
| Hanna and Donetto (2021) [45] | Understand more about the reproductive experiences of amputee women who are living with amputation | Global | 6 | All female. |
| Hansen et al. (2018) [46] | Examine the process of becoming a user of a transfemoral osseointegrated prosthesis, from the beginning of rehabilitation (after second stage surgery) and forward, as seen from the user's perspective. | Denmark | 7 | 2 females. Cause: 4 trauma, 3 tumour, Level: All TFA. Age range 37–70 yrs |
| Heavey 2018 [47] | Use a case study approach for analysing space as a narrative resource in stories about illness and recovery | UK | 1 | All Female, Cause: Diabetic dysvascular. Level: TFA. Age 60 yrs, Time since amp 50 yrs |
| Horne and Paul (2019) [48] | Understand the subjective experiences with chronic amputation pain and responses from family members, friends, and health care providers | USA | 11 | 5 females. Mean age 60.82 yrs |
| Jarnhammer et al. (2018) [49] | Explore experiences of persons in Nepal using lower-limb prostheses | Nepal | 16 | 6 females. Cause: 11 trauma, 2 infection, 2 tumour, 1 other. Level: 11 TTA, 1 KDA, 4 TFA. Mean age 38 yrs. Age range 21–67 yrs. Time since amp mean 10 yrs |
| Jeppsen et al. (2019) [50] | To better understand the resilience among Veterans who experienced combat-related amputations. | USA | 6 | Cause: 6 trauma |

(*Continued*)

**Table 6.** (*Continued*)

| Study | Aim | Location | Sample size (n) | Sample characteristics |
|---|---|---|---|---|
| Keeves et al. (2022) [51] | Explore the barriers and facilitators experienced by people with lower limb loss following a traumatic amputation that influence social and community participation between 18months and 5-years post amputation. | Australia | 9 | 2 females. Cause: 9 trauma. Level: 5 TTA, 4 TFA. Median age 59 yrs. Age range 50–64. Mean time since amp 35 months |
| Kim et al. (2021) [52] | Explore lived experiences, and identify common themes as well as vocabulary associated with fall-related events in LLP users | USA | 25 | 9 females. Cause: 14 trauma, 6 diabetic dysvascular, 3 infection, 1 tumour, 1 other. Level: 2 symes, 14 TTA, 1 KDA, 9 TFA, 4 bilat. Mean age 59.6 yrs. Age range 25–81 yrs. Time since amp 1–51 yrs |
| Koszalinksi and Locsin (2015) [53] | Describe the meaning of the experience of persons being cared for with prosthetic devices after lower limb amputation | USA | 12 | Unknown |
| Lee et al. (2022) [54] | explore the experience of self-managing after limb loss/ limb difference from the perspective of prosthesis users, prosthetists, and physical therapists. | USA | 10 | 6 females. Cause: 4 trauma, 2 diabetic dysvascular, 1 cancer, 3 congenital. Level: 5 TTA, 5TFA. Mean age 53.1 yrs. Mean time since amp 25.7 yrs |
| Lee et al. (2022) [55] | Examine the effects of the COVID-19 pandemic on physical activity levels in persons with limb loss | USA | 13 | Not known |
| Lehavot et al. (2022) [56] | Understand the experience of female veterans with prosthetic care and their prosthesis to inform direction of future research and clinical practice | USA | 30 | All female. Cause: 11 trauma, 9 diabetic dysvascular, 7 infection, 3 other. Level: 14 TTA, 15 TFA, 1 bilat. |
| Mathias and Harcourt (2014) [57] | Gain an in-depth understanding of the experiences and emotional responses of women with below-knee amputations to dating and intimate relationships | Jamaica, Columbia and USA | 4 | All female. Cause: 3 Trauma, 1 cancer. Level: All TTA. Age 18–29 yrs |
| Mattick et al. (2022) [58] | explore the factors influencing motivation of lower limb amputees engaging with prosthesis services in Mombasa, Kenya | Kenya | 10 | 2 females. Cause: 7 trauma, 3 diabetic dysvascular. Level: 10 TTA. Mean age 39 yrs. Age range 24–60 yrs. Time since amp 2–25 yrs. |
| Mayo et al. (2022) [59] | Interview persons with LEA about their mental health needs and to gauge their attitudes towards iCBT and/or online mental health supports | Canada | 10 | 1 female. Cause: 3 trauma, 6 diabetic dysvascular, 1 infection. Level: 7 TTA, 1 TFA, 1 bilat. Mean age 55.6 yrs. Age range 43–77 yrs. |
| McDonald et al. (2018) [13] | Explore outcomes that matter to prosthesis users who have experience using two different types of prosthetic feet | USA | 5 | 1 female. Cause: 2 Trauma, 1 diabetic dysvascular, 2 Infection. Level: All TTA, 2 Bilat. Mean age 45.6 yrs. Age range 41–59 yrs. Time since amp 2.7–14.5 yrs |
| Miller et al. (2020) [60] | Describe resilience characteristics meaningful to people with TTA in middle age or later, who use a prosthesis | USA | 18 | 3 females. Cause: 13 diabetic. Level: TTA. Mean age 60 yrs. Mean months since amp 60 yrs |
| Miller (2021) [61] | To identify psychosocial factors with potential to influence clinically relevant measures of physical activity, physical function, and disability in light of participants' narratives | USA | 20 | 2 females. Cause: All diabetic dysvascular. Level: 15 TTA, 2 TFA, 3 bilat. Mean age 63.4 yrs. Mean time since amp 5.5 yrs |
| Morgan et al. (2020) [62] | to evaluate an existing conceptual measurement model of mobility and identify high-level activity item content to include in an expanded PLUS-M item bank | USA | 29 | 6 females. Cause: 20 trauma, 2 diabetic dysvascular, 2 infection, 3 cancer, 2 other. Level: 23 TTA, 6 TFA, 4 bilat. Age range 25–74. Time since amp 0.9–49.8 yrs |
| Norlyk et al. (2016) [63] | Explore the lived experience of becoming a prosthetic user as seen from the perspective of persons who have lost a leg. | Denmark | 8 | 2 females. Cause: 2 trauma, 5 diabetic dysvascular, 1 infection. Level: 9 symes, 4 TTA, 4 TFA. Age range: 33–74 yrs |
| Poonsiri et al. (2020) [64] | Explore consumer satisfaction with prosthetic sports feet and the relative importance of different dimensions regarding prosthetic sports feet | Netherlands | 16 | 6 females. Cause: 5 trauma, 2 diabetic dysvascular, 2 infection, 5 tumour, 2 other. Level: 8 TTA, 6 KDA, 2 TFA. Mean age 37.5 yrs |
| Roberts et al. (2021) [65] | Gain an indepth understanding of prosthesis use from the perspectives of individuals with major LLAs | Canada | 10 | 4 females. Level: 5 TTA, 1 KDA, 4 TFA. Mean age 63.3 yrs. Age range 47–78 yrs. |
| Stucky et al. (2020) [66] | Explore the lived experience of people in Bangladesh following LLA and prosthetic rehabilitation to understand the facilitators and barriers to their work participation | Bangladesh | 10 | 3 females. Cause: 9 trauma, 1 diabetic dysvascular. Level: 7 TTA, 3 TFA, 1 bilat. Mean age 34.6 yrs, Age range 23–63 yrs |

**Table 6.** (Continued)

| Study | Aim | Location | Sample size (n) | Sample characteristics |
|---|---|---|---|---|
| Taylor (2020) [67] | Explore whether subjective statements, justifying a patient preference for microprocessor controlled prosthetic limbs over non-microprocessor controlled limbs, involves themes other than functional improvement | UK | 15 | All male. Cause: all trauma. Mean age 34.7 yrs. Age range 23–51 yrs |
| Turner et al. (2022) [68] | To understand the experiences of people with LLA during rehabilitation with a prosthesis | UK | 10 | 4 females. Cause: 3 trauma, 3 diabetic dysvascular, 1 cancer, 3 other. Level: 8 TTA, 1 TFA, 1 bilat. Mean age 53.7 yrs. Mean time since amp 6.53 yrs |
| Van Twillert et al. (2014) [69] | to provide a better understanding of the impact of barriers and facilitators on functional performance and participation and autonomy post-discharge | Netherlands | 13 | 4 female. Cause: 4 trauma, 8 diabetic dysvascular, 1 other. Level: 10 TTA, 2 KDA, 1 TFA. Age range 29–73 yrs. |
| Verschuren et al. (2014) [70] | to explore qualitatively how persons with a lower limb amputation describe and experience (changes in) sexual functioning and sexual wellbeing after LLA | Netherlands | 26 | 9 females. Cause: 7 trauma, 7 diabetic dysvascular, 4 infection, 6 cancer, 2 other. Level: 15 TTA, 1 KDA, 6 TFA, 2 HAD, 2 bilat. Mean age 47 yrs. Age Range 22–71 yrs |
| Wadey and Day (2018) [71] | To provide an original and rigorous account of Leisure Time Physical Activity among people with an amputation in England | UK | 22 | 14 females, mean age 42 yrs. mean time since amp 5 yrs |
| Ward Khan et al. (2021) [72] | To gain an in-depth understanding of women's experience of sexuality and body image following amputation of a lower limb to inform rehabilitation and clinical practice | Ireland | 9 | All females. Cause: 2 trauma, 5 diabetic dysvascular, 1 cancer, 1 other. Level: 6 TTA, 2 TFA, 1 Pelvic. Age range 35–62 yrs. Time since amp 1.5–31 yrs |
| Young et al. (2022) [73] | understand current and former military experiences when using MPK primary and backup devices with a view to helping guide decisions related to policy and potentially improve rehabilitation services | Canada | 6 | 0 females. Cause: 5 Trauma, 1 diabetic dysvascular. Level: 2 KDA, 4 TFA. Mean age 44.6 yrs. |

Cause = Cause of amputation. Level = Level of amputation. TTA = Transtibial amputation, KDA = Knee Disarticulation Amputation, TFA = Transfemoral Amputation, HAD = Hip Disarticulation Amputation. Bilat = Bilateral amputation.

*"And if we go for a walk, I'm able to hold my wife's hand. I haven't been able to do that for eight to ten years. Some people might think that isn't a big deal, but to me it means a lot." (Jon)* (Hansen et al., 2019) [46]

*About 4 months later after my amputation we actually went to the park and slid down the slide with her, swung on the swing, and ran around the park. I don't even want to think about my life without doing that."* (Crawford et al., 2016) [40]

Fulfilling previous roles appeared to create a sense of normality for participants as well as promoting adjustment by building confidence and self-worth.

*feeling responsible for the household allowed her amputation not to matter, allowing her to go on with life despite her altered body.* (Ward Khan et al., 2021) [72]

*Being mobile in their communities enabled participants to actively participate in society: . . . I am the secretary of the ward committee. I meet in disability forums and write minutes . . . I go on my own.* (Ennion and Manig, 2019) [43]

*Domain 2—I can participate in my important activities in the way I want to.* A subtheme of the second domain describing *how* people with limb loss want to participate was modified

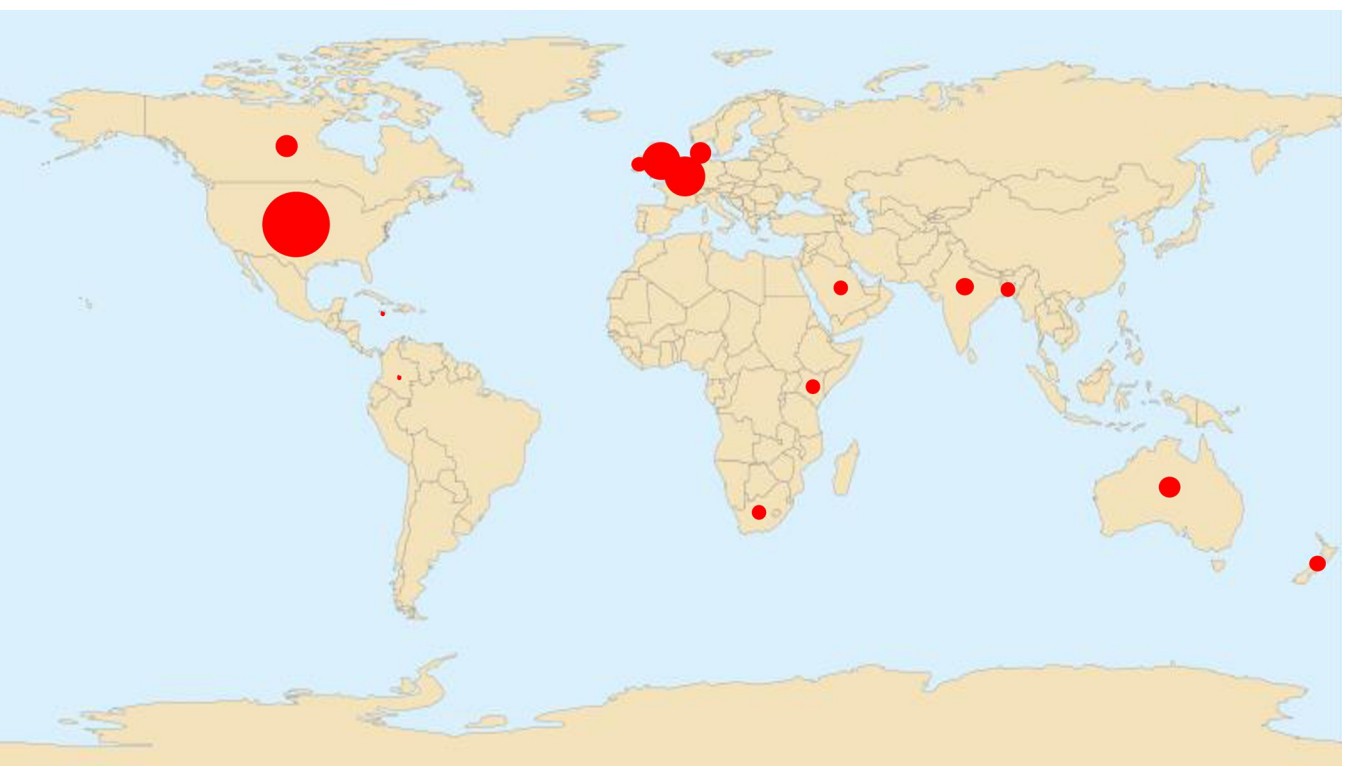

**Fig 2. Map of the world illustrating the geographical spread of participants involved in the included studies.**

from 'being able to do my activities easily' to reflect being able to do activities 'easily *and* well' (Table 10).

*Subtheme 2.2—Doing my activities easily and well.* Data describing participation in sport [13,37,64] raised the issue of doing an activity well. Participants described the need to perform well during sport to be competitive, not performing well could lead to reduced participation.

*Now, if I swim, the speed is gone and you always have a disadvantage. . . swimming is not what it used to be, all elderly swim faster than me. . .. . .I stopped with it. . ."* (Bragaru et al., 2013) [37]

**Domain 3 –My prosthesis works for me.** Originally domain three described the need for a prosthesis that is comfortable and easy to use. This domain was well supported by the data (Table 8); however additional data went beyond describing the comfort and ease of use i.e., burdensomeness of the weight, fit and suspension of the prosthesis, and also described the importance of the functionality of prosthetic componentry, i.e., prosthetic knees and feet, in enabling valued activities. This led to the domain being restructured into three subthemes (Table 11). The first two subthemes describe the original domain of 'comfort and ease of use' but have been presented separately as 'My prosthesis is Comfortable' and 'My prosthesis is easy to use' to reflect the importance of these individual aspects of the prosthesis, as described in the data. An additional third subtheme has been developed describing the importance of prosthetic componentry which enables participation.

**Subtheme 3.3—My prosthesis enables me to participate.** The function of prosthetic components and how they enable people to participate was described in 18 studies [13,36,61–

**Table 7. Study design and critical appraisal of study quality using the CASP qualitative appraisal tool (Yes = Light grey (1), Can't Tell = Dark Grey (2), No = Black(3)).**

| Author | Methodology | Data Collection approach | Analysis approach | CASP tool Section A | | | | | CASP tool Section B | | | | Total Score |
|---|---|---|---|---|---|---|---|---|---|---|---|---|---|
| | | | | Was there a clear statement of the aims of the research? | Is a qualitative methodology appropriate? | Was the research design appropriate to address the aims of the research? | Was the recruitment strategy appropriate to the aims of the research? | Was the data collected in a way that addressed the research issue? | Has the relationship between researcher and participants been adequately considered? | Have ethical issues been taken into consideration? | Was the data analysis sufficiently rigorous? | Is there a clear statement of findings? | |
| Day et al. (2019) [41] | Qualitative exploratory | FG | Inductive TA | | | | | | | | | | 9 |
| Hansen at al. (2019) [46] | Descriptive phenomenology | In depth Interview | RLW guiding principles | | | | | | | | | | 9 |
| Mathias and Harcourt (2014) [57] | IPA | On-line SSI | IPA | | | | | | | | | | 9 |
| McDonald et al. (2018) [13] | IPA | FG | IPA and adapted GT | | | | | | | | | | 9 |
| Morgan et al. (2020) [62] | Not stated | FG | TA | | | | | | | | | | 9 |
| Wadey and Day (2018) [71] | Longitudinal qualitative | FG, Obs, unstructured and SSI | Inductive TA | | | | | | | | | | 9 |
| Mattick et al. (2021) [58] | Qualitative | SSI | TA | | | | | | | | | | 9 |
| Norlyk et al. (2016) [63] | Phenomenology RLR | Longitudinal interviews | Thematic RLR | | | | | | | | | | 10 |
| Batten et al. (2020) [36] | Not stated | FG | Content / TA | | | | | | | | | | 10 |
| Devan et al. (2015) [42] | Qualitative | FG | General Inductive approach | | | | | | | | | | 10 |
| Kim et al. (2021) [52] | Qualitative | FG | Adapted GT | | | | | | | | | | 10 |
| Ward Khan at al. (2021) [72] | IPA | SSI | IPA | | | | | | | | | | 10 |
| Stuckey et al. (2020) [66] | Not stated | SSI | TA | | | | | | | | | | 11 |
| Bragaru et al. (2013) [37] | Phenomenology | SSI | Thematic codebook | | | | | | | | | | 11 |
| Camacho et al. (2021) [38] | Phenomenology | SSI | TA | | | | | | | | | | 11 |
| Hanna and Donetto (2021) [45] | Qualitative exploratory | Online posts | TA | | | | | | | | | | 11 |

(*Continued*)

**Table 7.** (Continued)

| Author | Methodology | Data Collection approach | Analysis approach | CASP tool Section A | | | | | | CASP tool Section B | | | Total Score |
|---|---|---|---|---|---|---|---|---|---|---|---|---|---|
| | | | | Was there a clear statement of the aims of the research? | Is a qualitative methodology appropriate? | Was the research design appropriate to address the aims of the research? | Was the recruitment strategy appropriate to the aims of the research? | Was the data collected in a way that addressed the research issue? | Has the relationship between researcher and participants been adequately considered? | Have ethical issues been taken into consideration? | Was the data analysis sufficiently rigorous? | Is there a clear statement of findings? | |
| Horne and Paul (2019) [48] | Empirical Phenomenology | SSI | Empirical Phenomenology | | | | | | | | | | 11 |
| Keeves et al (2022) [51] | Qualitative exploratory | SSI | TA | | | | | | | | | | 11 |
| Miller et al. (2020) [60] | Mixed methods | SSI | Directed content analysis | | | | | | | | | | 12 |
| Miller (2021) [61] | phenomenology | SSI | Six step method by Cresswell | | | | | | | | | | 12 |
| Hafner et al. (2016) [44] | Not stated | FG | TA | | | | | | | | | | 13 |
| Jeppsen et al. (2019) [50] | conceptual framework | SSI | Content | | | | | | | | | | 13 |
| Verschuren et al. (2015) [70] | Descriptive qualitative approach | SSI | TA | | | | | | | | | | 13 |
| Lee et al. (2022) [54] | Qualitative | Interviews | Constant comparison method | | | | | | | | | | 13 |
| Lehavot et al. (2022) [56] | Qualitative | SSI | content analysis | | | | | | | | | | 13 |
| Poonsiri et al. (2020) [64] | Mixed methods | SSI | Qualitative interpretation | | | | | | | | | | 14 |
| Taylor (2020) [67] | Pilot | Written statements | TA | | | | | | | | | | 14 |
| Jarnhammer at al. (2018) [49] | Qualitative | SSI | Content | | | | | | | | | | 14 |
| Koszalinski and Locsin (2015) [53] | HPA | SSI | HPA | | | | | | | | | | 15 |
| Christensen et al. (2017) [39] | Qualitative exploratory | SSI and obs | Inductive latent TA | | | | | | | | | | 15 |
| Mayo at al. (2022) [59] | CFIR | Interviews | Codebook TA | | | | | | | | | | 15 |
| Young et al. (2021) [73] | IPA | SSI | IPA | | | | | | | | | | 15 |

*(Continued)*

**Table 7.** (Continued)

| Author | Methodology | Data Collection approach | Analysis approach | CASP tool Section A | | | | | | CASP tool Section B | | | Total Score |
|---|---|---|---|---|---|---|---|---|---|---|---|---|---|
| | | | | Was there a clear statement of the aims of the research? | Is a qualitative methodology appropriate? | Was the research design appropriate to address the aims of the research? | Was the recruitment strategy appropriate to the aims of the research? | Was the data collected in a way that addressed the research issue? | Has the relationship between researcher and participants been adequately considered? | Have ethical issues been taken into consideration? | Was the data analysis sufficiently rigorous? | Is there a clear statement of findings? | |
| Ennion and Manig (2019) [43] | Qualitative exploratory | SSI | TA | | | | | | | | | | 16 |
| Heavy (2018) [47] | Narrative | In depth Interview | Unknown | | | | | | | | | | 16 |
| Van Twillert et al. (2014) [69] | Mixed methods | SSI | Framework analysis using ICF | | | | | | | | | | 16 |
| Abouammoh et al. (2021) [35] | Phenomenology | FG and SSI | TA | | | | | | | | | | 17 |
| Roberts et al. (2021) [65] | Qualitative descriptive | SSI | TA | | | | | | | | | | 17 |
| Crawford et al. (2016) [40] | Observational qualitative | SSI | TA | | | | | | | | | | 18 |
| Lee et al. (2022) [55] | Mixed methods | SSI | Constant comparison method | | | | | | | | | | 22 |
| Turner et al. (2022) [68] | Mixed methods | SSI | TA | | | | | | | | | | 22 |

IPA–Interpretive phenomenological approach, RLW- Reflective life world research, GT–Grounded theory, TA- Thematic analysis, HPA–Hermeneutic phenomenological approach, CFIR—Consolidated Framework for Implementation Research, SSI–Semi structured Interview, FG–Focus Group, Obs–Observations.

**Table 8. Examples of how data from the qualitative synthesis support the pre-existing framework domains.**

| Pre-existing framework domains and subthemes | Examples from qualitative synthesis data |
|---|---|
| **Domain 1—I am able to participate in my important activities** | |
| **Subtheme 1.1 -** **Walking again** (10 papers—[13,35,36,40,44,53,58,63,65,73]) | *I just wanted to get prosthesis and be able to walk again, those were my expectation, I had been told that there are false legs that one can get and they help one to be able to walk. (Juma)* (Merrick et al., 2022 [76]) *I would be tempted to add something to [a definition of mobility] about the ability to accomplish wanted or needed tasks.PT2.6 (Hafner et al., 2016 [44])* |
| **1.2—Important activities at home** (11 papers–[36,42,73,44,49,53,63,65–67,69]) | *This leg [prosthesis] has managed to help me a lot; because of this leg [prosthesis] I'm able to do work, go to the toilet and carry things around my house, and I can travel and walk. (Female N, living in urban area) (Jarnhammer et al., 2018 [49])* *For most participants, the prosthesis was actively used to complete activities of daily living such as cooking, cleaning, and laundry. One individual explained, "Well, because it's a pain in the ass trying to cook it all from your wheelchair, especially in front of the stove. Right, because I'm terrified of something like the pot tipping over, whatever and scalding me." (Participant 08)* (Roberts et al., 2021 [65]) |
| **1.3—Important activities in my community** (24 papers [13,35–37,40,43,44,46,49–53,55,58,59,62,63,65–67,69,71,73]) | *Expanded mobility, perhaps the most important and commonly reported outcome for study participants, was experienced in a unique way for each individual person and his or her lifestyle. (McDonald et al., 2018 [13])* *Another participant expressed immense satisfaction that her prosthesis allowed her to engage in physical activity with her family, "We do a lot of swimming, we do badminton, we played as a team outside of the house, I play volleyball. . ." (Participant 14)* (Roberts et al., 2021 [65]) *One example was characteristics of the terrain, such as sand or uneven terrain: "I cried the first time I was on sand. I thought I would never be able to walk on sand again. I had to leave the beach" (PT1.3). (Morgan et al., 2020 [62])* *Another participant had changed work roles as she was unable to walk the distances required to be a professional cleaner. One participant was unable to walk the required distance to public transport for work. (Batten et al., 2020 [36])* |
| **Domain 2—I can participate in my important activities in the *way* I want to** | |
| **2.1—Doing my activities independently** (14 papers [13,35,41,44,46,48,49,51,53,58,62,63,65,66]) | *Since receiving the prosthetics, he had opened his own barber shop and how he could now "depend on myself." He was not alone; others spoke significantly about reduced dependency: I can take myself to the shop without any help, unlike there before when I used to depend on people to help me, I can go by myself to the toilet without asking for help (Mohamed) (Mattick et al., 2022 [58])* *There was a reluctance to ask for support from their spouse or extended family–to not to be a burden; particularly given that failure to fulfil an expected role left some women feeling their spouse may leave: If I ask someone [to help] it might be hard for them too. It becomes very difficult for my mother and sister-in-law when I go home. When I am dependent on them, this actually increases their workload.–Fatima (F, 23)* (Stuckey et al., 2020 [66]) |

*(Continued)*

Table 8. (Continued)

| Pre-existing framework domains and subthemes | Examples from qualitative synthesis data |
|---|---|
| **2.2—Doing my activities easily**<br>(21 papers [13,36,37,40–42,44,46,47,53,60–64,66,67,69,71–73]) | *You need to create a day-to-day life, where you do not think so much about it anymore. It needs to become a routine that you need to put on a sock in the morning, and remember to wash it before going to bed in the evening. . . it should be like brushing your teeth. Something you do without even thinking about it. (Daniel, third interview) (*Norlyk et al., 2016 [63])<br>*Additionally, participants described some physically focused activities that were no longer possible after LLA because the effort, adaptation, and/or time were too great. For example, some participants reported that they no longer mow their lawn because the effort, adaptation, and time to push the lawn mower, maintain balance, and manage the associated tasks were too great following LLA. (*Miller 2021 [61]) |
| **2.3—Doing my activities without falling over**<br>(18 papers [13,36,37,42,44–46,51,52,54,60–62,64,66,68,71,73] Refs) | *I got frustrated when I had tripped and fallen multiple times with my mechanical knee. It's super frustrating, obviously. I want to be able to function and not to worry about falling, like anybody."*<br>(Young et al., 2022 [73])<br>"*I will go ahead and stop going to, you know, to the stores to pick up a bunch of little items because it's just not worth the hassle going by a slip hazard or a trip hazard or an ice patch, something like that." (Male, 59 years old, TT, 8 years since amputation)* (Kim et al., 2021 [52]) |
| **2.4—Doing my activities with as little equipment as possible**<br>(11 papers [36,38,72,44,45,52,53,58,63,65,66]) | *I dare not have too high hopes. . . But I do have a dream that I can walk down the street without a stick for support, that is a big dream (emphasis). . . and it would be a major victory for me to go shopping without anyone realizing that I walk with an artificial leg. (Hanna, third interview)* (Norylk et al., 2016 [63])<br>*Many participants implemented the use of mobility aids and seated rests to overcome challenges associated with community ambulation, such as unsteadiness or fatigue. One participant explained, "But, when I got my walker, I just turn it around backwards and I sit down and relax and get my breath and, get everything back to normal and then continue on." (Participant 06)*<br>(Roberts et al., 2021 [65]) |
| **Domain 3—My prosthesis is comfortable and easy to use**<br>(17 papers [36,40,45,46,49,51,52,54,56,58,61–63,65,68,69,73]) | *I think a lot of effort is put into the ankles and the legs. . .but I think actually the socket fit is something that's really important but of course not as glamorous and therefore gets forgotten." (Prosthesis User 1, Female individual)* (Turner et al., 2022 [68])<br>*There's been times where I've had. . . the occasional blister and because the stump is shrinking so much. . . it gets irritation on the side of the socket and then that becomes painful. That means you have to stay off your leg for a few days until the swelling goes down and then you can redo it all again. . . it's sort of hit and miss through the year, you never know when a blister is going to happen. [Rob_M_TKA_61–70_<3 years]* (Keeves et al., 2022 [51])<br>"*[Sweat] actually comes over the top of the liner. . .it's obviously quite uncomfortable and it can wet shorts and wet trousers because the sweat is actually coming over the top of the liner." (Prosthesis User 6, Male)* (Turner et al., 2022 [68])<br>*One participant had a suspension system that made it quick and easy to don, enabling walking. (*Batten et al., 2020 [36]) |

(*Continued*)

**Table 8.** (Continued)

| Pre-existing framework domains and subthemes | Examples from qualitative synthesis data |
|---|---|
| **Domains 4—If I have pain, I am able to manage it** (7 Papers [38,40–42,45,48,62]) | *While some participants reported that a pain-free day was possible, for most a good day involved better management of pain, allowing them to engage in activities that they wanted to do.* (Day et al., 2019 [41])<br><br>*For some participants, pain was a reason for non use of the prosthesis, "Some days I don't even put it on, don't even tell me to put it on, because I'll get mad at you. There's nothing worse than having a pain you can't control. You know, and the only way I can control it is to stay off both my feet." (Participant 02)* (Turner et al., 2022 [68])<br><br>*You know like part of the package when you got a limb you are going to get pain here and there. . ..Ah it is it can be really uncomfortable yeah, but you just got to sort of carry on through it. . . (Jack, Int 2)* (Devan et al., 2015 [42]) |
| **Domains 5—I am able to accept my new normal** | |
| **5.1—Chasing normality** (14 papers [35,37,39,41,42,46,48,53,57,58,61,63,67,71]) | *You're making me think. I don't know. It's a deep question. I haven't addressed it, even though I think I have. I haven't addressed the fact that I'm disabled. I've come to terms with it, I get on, but I probably haven't properly. I don't really like that word. What does it mean? I don't like it. It makes me different. I don't want to be different. I just want to be the same as everyone else. I just want to just fit in. To just be.* (Wadey and Day, 2018 [71])<br><br>*Depending on the degree of regained mobility the participants strived to re-conquer a daily life that resembled their previous lives.* (Norlyk et al., 2016 [63])<br><br>*A good day is when I just feel like everybody else. [Gloria] Moving away from the amputation.* (Day et al., 2019 [41]) |
| **5.2—Adjusting to limb loss** (32 papers [13,35–39,41,43,44,46–51,53,56–67,69–72]) | *"I know that life is worth living and there is still that out there, but it's hard to come back to that."* (Jeppsen et al., 2019 [50])<br><br>*Everybody's looking to the past, how they used to be. Uh. . .. So for me, you go to wedding and you see someone is dancing all night. I was that guy. Can I dance again all night? I had a tree at my cottage. A poplar, it was 40–50 feet tall. I climbed way over there with a chainsaw in my hands so I chop it down. Can I do that again? (ID#17, Male outpatient, 51, traumatic aetiology)* (Mayo et al., 2022 [59])<br><br>*"It doesn't matter how you do it because everybody has something, then you feel more at home and less stared at . . .. . . you feel less different. . ..and then you accept it"* (Bragaru et al., 2013 [37])<br><br>*For example, a participant stated he could not squat or be down on one knee to change a car tire and, "That's a limitation that I've adapted to. So, I just put a stool down and then sit on the stool, and then do what I gotta do. So, you just have to take the limitations, and then adapt to do things that way," (69 years old; 2.5 years post-TTA).* (Miller et al., 2020 [60])<br><br>*"Only the strong survive baby! If you don't adapt to the circumstances, my gosh, you are going to have a miserable life."* (Camacho et al., 2021 [38]) |

(*Continued*)

**Table 8.** (Continued)

| Pre-existing framework domains and subthemes | Examples from qualitative synthesis data |
|---|---|
| **5.3—Sense of achievement**<br>(13 papers [13,37,63,64,71,38–41,45,47,60,61]) | *Participants described pride in success, building their confidence in pursuit of challenging goals. Another participant stated, "[Being active] is incredibly gratifying. I mean, in this circumstance in particular, maybe because it's like I've been recovering something. That feeling like, yea. I mean, it makes me really proud," (54 years old; 1 years post-TTA). (Miller et al., 2020 [60])*<br>*Participants described personally meaningful goals and implementation of strategies, both successful and unsuccessful, to minimize identified barriers, achieve goals, and reduce their disability. (Miller et al., 2020 [60])* |

64,66,67,72,73,37,42–44,49,53,56,58]. Participants described wanting a leg that was waterproof so they could go fishing, or a flexible ankle so they could lift objects at work. A limb that did not enable function could prevent participation or make it more challenging [36,37,73,42,49,56,58,62,66,67,72].

> *Like the last time when I went to the Amputee Clinic, I said, 'I like to go fishing and I would like to go canoeing a little bit and stuff, but I can't get this prosthesis wet, is there a type of prosthesis I can get wet?* (Lehavot et al., 2022) [56]

> *There's a lot of lifting in my job and fitting and stretching, not having one of the ankles, you lose a lot of balance and so you do tend to use your back like a crane a lot more than that I did when I had two legs, just 'cause it doesn't, you haven't got the balance so you just, you find yourself by necessity bending when I know I should be bending from the knees but I can't get the lift off a prosthesis in the same way (Mitchell, FG3)* (Devan et al., 2015) [42]

Trust in the prosthesis not to give way underneath them or break also appeared to be an important factor in componentry enabling participation, particularly in relation to the prosthetic knee [13,44,61,62,64,73].

> *"It takes me a little bit to trust my leg that when I take a step, it is going to be there. I have had it break on me too. I have had to gain that trust with my leg then lost it, then gained it, then lost it. So over time it has been hard for me to really trust it. That when I take a step it's going to be there for me. It's not going to break. It's not going to send me flying"* (Morgan et al., 2020) [62]

> *participants identified the pervasive fear of falling as the major issue, as they did not trust the knee unit to appropriately respond and provide stability. This affected mood and willingness to engage in daily activities* (Young et al., 2021) [73]

**Table 9. Development of domain 1 –I am able to participate in my important activities and roles.**

| Pre-existing framework domain and subthemes | Newly expanded domain and subthemes |
|---|---|
| **Domain 1**—I am able to participate in my important activities | **Domain 1**—I am able to participate in my important activities **and roles** |
| 1.1 Walking again | 1.1 Walking again |
| 1.2 Important activities at home | 1.2 Important activities at home |
| 1.3 Important activities in my community | 1.3 Important activities in my community |
|  | **1.4 Fulfilment of roles** |

**Table 10. Development of domain 2 –I can participate in my important activities in the *way* I want to.**

| Pre-existing framework domain and subthemes | Newly expanded domain and subthemes |
|---|---|
| **Domain 2**—I can participate in my important activities in the *way* I want to | **Domain 2**—I can participate in my important activities in the *way* I want to |
| 2.1 Doing my activities independently | 2.1 Doing my activities independently |
| 2.2 Doing my activities easily | 2.2 Doing my activities easily **and well** |
| 2.3 Doing my activities without falling over | 2.3 Doing my activities without falling over |
| 2.4 Doing my activities with as little equipment as possible | 2.4 Doing my activities with as little equipment as possible |

Insufficient trust in prosthetic componentry was shown to prevent participation in important activities or require adaptation. Conversely trust appeared to inspire confidence in the limb, as well as individual capabilities.

> *If I feel like I can trust the leg or socket, then as far as being mobile, I feel like I can do. . . anything.*" (Hafner et al., 2016) [44]

**Domain 4 –If I have pain, I can manage it.** The analysis did not reveal any new information relevant to this domain.

**Domain 5 –I am able to accept my new normal.** Large amounts of the data from the included studies were mapped onto this domain which has been expanded and renamed in parts (Table 12). The subtheme 'Chasing normality' was renamed to 'Feeling a sense of normality' to better capture the domain as described by people with limb loss. The subtheme 'adjusting to limb loss' was expanded and is now presented in two subthemes highlighting the importance of 'adapting and accepting my limitations' and 'accepting my appearance'. An additional fifth subtheme was also identified describing lifelong health and wellbeing.

*Subtheme 5.2—Adapting and accepting my limitations.* Data from 14 studies supported this subtheme [36,38,43,47,48,51,58,60,62,65,66,69–71,]. Participants discussed the need to adjust to the changes they had experienced by adapting how they did their daily tasks.

> *For example, a participant stated he could not squat or be down on one knee to change a car tire and, "That's a limitation that I've adapted to. So, I just put a stool down and then sit on the stool, and then do what I gotta do. So, you just have to take the limitations, and then adapt to do things that way," (69 years old; 2.5 years post-TTA).* (Miller et al., 2020) [60]

Some participants described these adaptations as frustrating and indicative of the lives they had lost.

> *It's hard I guess you have to think about things a little bit differently. How you do things, takes a bit longer to do. . . and that sort of thing which is a bit frustrating. . . you knew what*

**Table 11. Development of domain 3 –My prosthesis works for me.**

| Pre-existing framework domain and subthemes | Newly expanded domain and subthemes |
|---|---|
| **Domain 3**—My prosthesis is comfortable and easy to use | **Domain 3—My prosthesis works for me** |
| | **3.1 My prosthesis is comfortable** |
| | **3.2 My prosthesis is easy to use** |
| | **3.3 My prosthesis enables me to participate** |

**Table 12. Development of domain 5—I am able to accept my new normal.**

| Pre-existing framework domain and subthemes | Newly expanded domain and subthemes |
|---|---|
| **Domain 5**—I am able to accept my new normal | **Domain 5**—I am able to accept my new normal |
| 5.1 Chasing normality | **5.1 Feeling a sense of normality** |
| 5.2 Adjusting to limb loss | **5.2 Adapting and accepting my limitations** |
| | **5.3 Accepting my appearance** |
| 5.3 Sense of achievement | 5.4 Sense of achievement |
| | **5.5 Lifelong health and wellbeing** |

*you could do before and you're never going to achieve that again now. (Male, Transfemoral, 35–50 yrs old)* (Keeves et al., 2022) [51]

However, participants appeared to view success as accepting what they could no longer do and focusing on what they could do. This seemed to be enabled by a problem-solving attitude and engendered a sense of pride in achievements.

*There are just so many more possibilities than you ever thought there would be. I can't do this, I can't do that. You spent so much time trying to get back to who you were, and this event says, you may be not be able to get back to who you were but look at all these amazing things you can do and can go on to achieve. It opens the gate to any other ideas you had in mind that you thought you couldn't do; it's just amazing. You realise that you are capable of so much more than you thought you were.* (Wadey and Day, 2018) [71]

*Subtheme 5.3—Accepting my appearance.* Data from 23 of the studies focused on the importance of adjusting to an altered appearance following amputation [35,36,37,39,41,49–51,53,56–58,60–62,64–67,70–72], both in terms of how participants saw themselves, and how they perceived others saw them.

*"I admit that I wanted to quit studying at the university many times due to that feeling I had. Even if I tried to convince myself to live with my new different look peacefully and accept my new self. . .I am in a constant battle from the inside." (female, 26 years).* (Abouammoh et al., 2021) [35]

This was often described in scenarios where staring or comments from others may have reinforced a negative self-image [37,72]. However, interaction with others was also described positively in accounts of acceptance from others leading to greater self-acceptance [41,57,65,70].

*Witnessing someone else's acceptance of the prosthesis that they themselves had sometimes struggled with helped them to feel understood and accepted for who they were* (Mathias and Harcourt, 2014) [57]

This interaction was also described in reverse with greater self-acceptance appearing to result in improved interactions with others [41,57].

*Samantha reflects on how her own growing sense of comfort about the prosthesis had had a positive impact on the reactions of others, which in turn had increased her sense of confidence*

*further: Once I was comfortable with it, everyone around seemed to be. . .. (Samantha)* (Mathias and Harcourt, 2014) [57]

Ultimately these experiences of acceptance were viewed positively and indicate the importance of being able to address issues of appearance during rehabilitation and recovery. Some participants described using clothing for concealment purposes to manage concerns about appearance. However, clothing also appeared to contribute to concerns, especially in certain social situations [46,61,62,75,77].

*I suppose it's a female thing but if you are invited somewhere and it's a posh do and you're getting dressed up and then you look down at your shoes. And then it's like bloody hell, from here [head] to here [knee] I look ok, and then I have a pair of trainers on my feet.* (Carly) (Day et al., 2019) [41]

Other facilitators of acceptance, concerning both appearance and function, were described as a positive problem-solving attitude [13,41,49,53,55,56,58,62,63,65,66,70,74], being able to participate in important activities and roles [42,44–46,51,64,70,71,76], time since amputation [62], spirituality [40,53,63,71] and peer support [13,40–42,44–46,50,54,58,67,76,77].

*"It doesn't matter how you do it because everybody [peers] has something, then you feel more at home and less stared at . . . . . . you feel less different. . ..and then you accept it* (Bragaru et al., 2013) [37]

*Subtheme 5.5—Lifelong health and wellbeing.* Participants in 9 of the included studies highlighted concerns about the impact amputation and prosthesis use would have on their health and wellbeing throughout their life course [37,40,42,45,57,61,62,71,72]. Participants described concerns about the impact of amputation on their remaining joints [62], the need to remain physically active to avoid health issues later in life, and to manage weight gain [37,40,57].

*I think just talking about hopping. I have an example of what happens to you 20 years later. I had really bad arthritis in my knee. I have torn my ACL and if I had not [hopped on one leg] growing up, it probably would be better.* (Morgan et al., 2020) [62]

*For the ones who stated that they cannot live without it, "sport is more a necessity" and, even if it was "not perceived as a fun activity", the individual still participated in sports because otherwise he or she had the feeling that it would have negative consequences for his or her health.* (Bragaru et al., 2013) [37]

**Interconnected nature of outcome domains.** Data from the qualitative synthesis demonstrated that outcome domains of importance are interconnected, which was first introduced in our qualitative paper developing the original conceptual model [20]. Many examples were presented of how the different domains interacted, for example, how socket comfort issues prevented participation which in turn impacted adjustment and mental wellbeing, or how a lack of trust in the prosthesis caused a fear of falling, which led to reduced community participation. This analysis concurs that a successful outcome appears to be multi-faceted and requires a multi-domain measurement approach, if the outcome of prosthetic rehabilitation is to be captured in a holistic, meaningful way. Fig 3 visualises the expanded 'ECLIPSE' model, and the interconnected nature of the domains of importance.

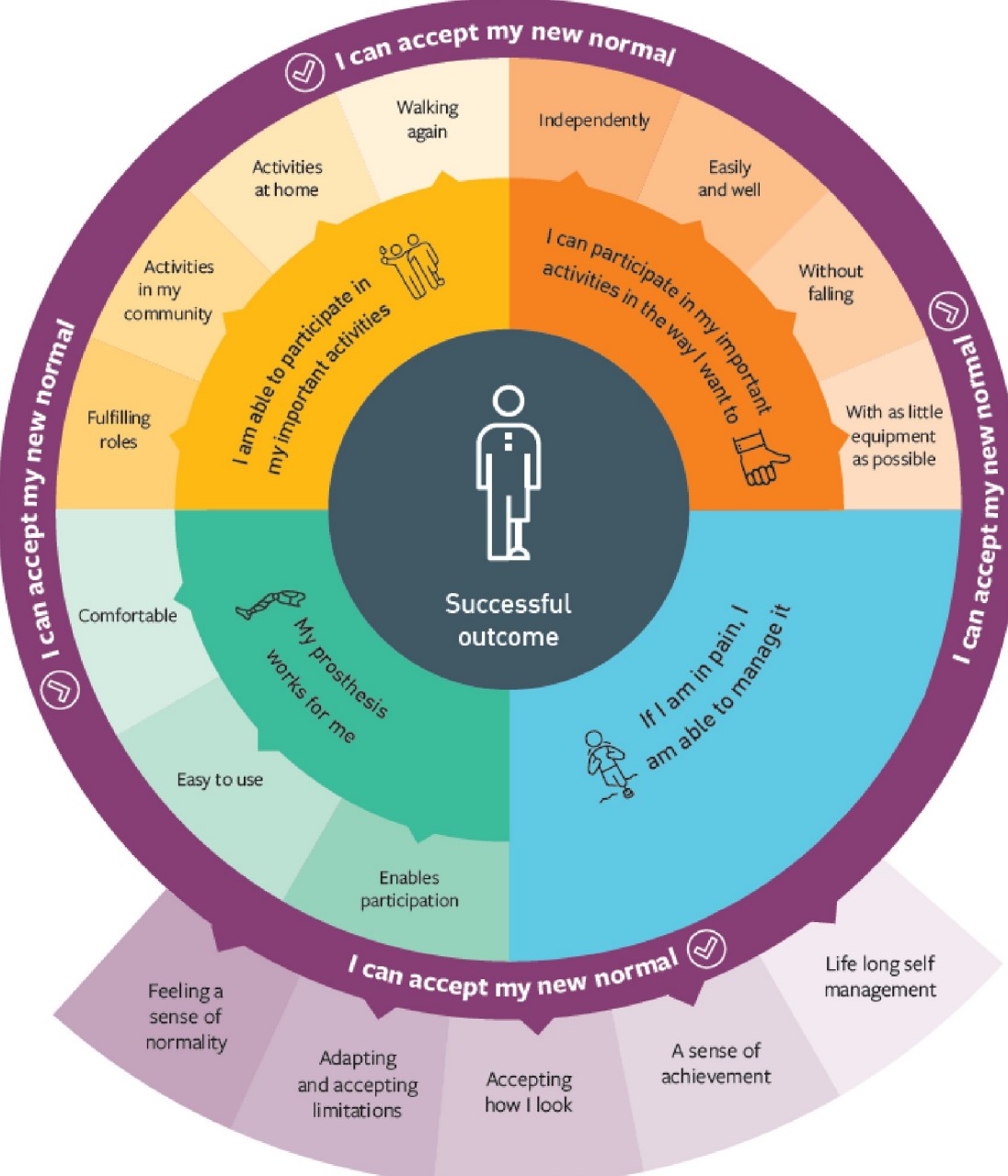

**Fig 3. Expanded conceptual model–the ECLIPSE model of meaningful outcome domains of lower limb prosthetic rehabilitation.**

## Discussion

This review presents a modified and enhanced conceptual model of outcome domains of importance following lower limb prosthetic rehabilitation from the perspective of people with limb loss. Having been initially developed during a primary qualitative inquiry with 37 prosthetic users [20], it has now been rigorously examined in this systematic review using data from 40 papers describing the experiences of 539 lower limb prosthetic users from a variety of settings. The application of 'best fit' framework synthesis allowed us to re-examine and review

domains of importance in the context of the lower limb loss literature and produce a second iteration, now named the ECLIPSE model, which more comprehensively attempts to describe this phenomenon.

The systematic review demonstrated that many of the original domains in the pre-existing model were supported by data from the literature. Thus, the model illustrates the importance of domains such as being able to participate in meaningful activities in a way individuals are happy with, having a comfortable and easy to use prosthesis, being able to manage pain, and acceptance of the new normal. However, our understanding of these concepts has been deepened during this synthesis and has led to several of the domains being expanded and re-specified.

The first domain of the ECLIPSE model described the ability to participate in important activities following prosthetic rehabilitation. Data from the review identified that following limb loss people also appeared to value being able to return to important roles. Role fulfilment was often described alongside valued activities, with the valued activity appearing to gain additional meaning when participation enabled a return to previously valued roles. This phenomenon has also been described following traumatic brain injury [77], stroke [78] and during older persons rehabilitation [79]. A meta synthesis of studies exploring experiences of recovery following traumatic brain injury reported that returning to valued roles had a significant impact on individuals' self-worth and that without access to these roles people struggled to define their sense of self-identity [77]. Participation in valued roles following limb loss has been described as contributing to an individual's sense of self-identity, which can be significantly disrupted by the amputation [80,81]. However, a previous review of psychosocial adjustment to amputation suggested that successful recovery involves individuals *adapting* to changes in roles, alongside functioning and body image, and incorporating these changes into a new self-identity [81].

This review also demonstrated the importance of the right prosthetic componentry as an outcome domain of importance and led to re-specification of domain three (My Prosthesis works for me). What appeared to define 'the right' componentry (i.e., prosthetic knee, foot, suspension system etc.) was its ability to enable participation in important activities and roles, i.e., waterproofing to enable fishing, or a stable 'trusted' knee for walking on uneven ground. This has been reported in qualitative studies by Liu et al. [82] and Murray [83] who describe the prosthesis as key to enabling valued activities. Many different prosthetic components, designed to meet the varied functional needs of limb wearers are currently available [84], nonetheless, it may be challenging to identify a product that enables *all* the different activities people engage in. Having multiple prostheses for different activities could be a solution, i.e., a cycling leg or special occasion leg. However, this may be limited by financial constraints or prosthetic service provision, and may not reflect the way people often transition seamlessly between activities throughout the day. The importance of prosthesis functionality, as well as the addition of role fulfilment to domain one (I am able to participate in my important activities and roles), highlights the need for considered discussion between patients and healthcare professionals to clearly define what activities and roles are *most* important, and how these can be enabled through prosthetic prescription and rehabilitation. It may also be important to discuss what functionality might be lost as prescriptions change across the life course, and how this affects participation. This patient-centred approach emphasises the need for multidisciplinary input, especially considering the role of the Occupational Therapist, both during rehabilitation and lifelong prosthetic care, in order adopt an ongoing focus on participation. This focus may also challenge the current approach to outcome measurement, where tools identify the activities included in the assessment, such as walking in a crowded shopping centre or visiting a friend's house. Meaningful outcome measurement may require tools that allow patients

to define what activities are most important to them as an individual, and therefore should be captured as a measure of success.

Domain five of the ECLIPSE model, 'I am able to accept my new normal', was revised most significantly during this review, with three new subthemes created. This may be due to the nature of qualitative research which focuses on views and experiences and is often used to explore adjustment following amputation. The first new subtheme, 'being able to adapt to and accept my limitations' appears to be a common theme described in the rehabilitation literature characterising recovery from trauma or the management of long term conditions, i.e., anterior cruciate reconstruction [85], Parkinson's Disease [86] and traumatic brain injury [87]. A study by Rosengren et al., [86] exploring the experiences of patients with Parkinson's disease found that greater life satisfaction is achievable as people adapt to their new life situation, which involves a process of transition often based on acceptance.

This review also highlighted the need to adjust to an altered appearance following amputation. The wealth of literature describing this outcome led to its creation as a new subtheme and appeared to suggest that individuals need to adjust to how they see themselves, as well as their perception of how others see them, and that these experiences are intricately linked. This is supported by Cooley's 'Looking-Glass self' theory [88] which describes how individuals base their sense of self on how they perceive others view them. The importance of adjusting to an altered appearance following limb loss is described in several studies included in a qualitative meta synthesis by Murray and Forshaw [89]. They describe the importance of using the prosthesis to moderate the reaction of others and conceal limb loss. They also highlight that adjustment to changes in self-image appear to occur over extended periods of time as people learn to accept the limitations of the prosthesis.

Both of the subthemes, 'adjusting and accepting my limitations' and 'accepting my appearance', as well as the final subtheme describing lifelong health and wellbeing, indicate the importance of both physical and psychosocial recovery following lower limb amputation. Rehabilitation programmes may need to address both aspects in an integrated way to provide holistic patient-centred care. However, it is clear interventions may not only be required in the immediate post amputation period, and that ongoing physical and psychosocial support may be crucial to address changing lifelong needs.

The interconnected nature of outcome domains of importance, first documented in the authors' primary qualitative work [20] and supported by this review, suggests the need for a multidomain approach to outcome measurement in prosthetic rehabilitation. Many examples of how domains may influence each other were described in this analysis, for example socket comfort issues leading to reduced participation in important activities. Although the findings presented here, and visualised in the ECLIPSE model (Fig 3), recognise the interconnected nature of domains of importance and the need to measure them in a holistic way to capture meaningful success, further research is needed to understand the nature of the relationships between domains.

The ECLIPSE model presents a patient-centred representation of outcome domains of importance following lower limb prosthetic rehabilitation. The model could be used to direct the course of rehabilitation and highlights the need for physical and psychosocial interventions. Although several professional networks have published prosthetic rehabilitation guidelines [5,15,90,91], none include the views of patients, and no guidance is available to inform psychosocial management. Despite many papers describing the psychosocial impact of amputation [89], little research has been undertaken to evidence treatment options. Future work may be needed in order to understand how the domain of 'accepting my new normal' might be addressed during prosthetic rehabilitation.

The ECLIPSE model also provides guidance for which domains may be most important to measure following prosthetic rehabilitation, or in research, and could underpin a future Core Outcome Set. However, given the previously described challenges of meaningful patient involvement in COS development [12], care needs to be taken that the contribution of wider stakeholders in the COS process does not diminish the voice of prosthetic users themselves. The OMERACT initiative [92] which develop COS' for Rheumatoid arthritis have acknowledged this concern and developed a patient COS which explicitly acknowledges that what is important to patients may be different and in need of specific consideration [93]. The ECLISPSE model could represent a patient Core Outcome Set, informing measurement in both research and clinical practice, and ensuring a person-centred focus. Future work is required to identify outcome measurement tools which capture these domains.

The design and quality of studies included in this review varied considerably. Critical appraisal using the CASP tool was undertaken to summarise key quality issues and provide some context to the overall findings of the review but was not used to exclude studies or indicate strength of findings. The usefulness of critical appraisal is debated in the literature due to the variation in appraisal decisions between reviewers experienced in qualitative research reported when using the same and different appraisal tools, or solely based on their independent judgement [94]. The impact of including or excluding low quality studies on the findings of a review has also been found to have little impact [95] and this is why no studies were excluded based on quality alone in this review.

Within this review the key quality issue identified in 28 of the 40 studies was undue consideration of the influence of the researcher on the research process, which could impact the dependability and confirmability of this reviews findings [96], and is considered an area of concern for qualitative research in the field of prosthetic rehabilitation. In light of this, data included consisted only of first-person quotations, or interpretations that were directly supported by first-person quotations, in an attempt to ground the findings in the experiences of participants [97].

A further quality issue in 17 studies was insufficient information about whether recruited participants were best placed to answer the research question. However, data describing the study sample characteristics was presented in 34 of the studies allowing transferability to be considered. This review captures the experiences of a large sample (n = 539) of lower limb prosthetic users living in 15 different countries. Views and experiences from participants with different levels of amputation, a variety of causes and a wide age range were included, representing a varied sample capturing many different voices. However, despite the range of study settings, 90.2% of participants live in high-income countries. Far fewer qualitative studies have been undertaken exploring the lived experiences of lower limb prosthetic users living in low- and middle-income countries (LMICs). Due to limited representation of these individuals, it is unclear whether these findings are transferrable and whether the ECLIPSE model describes outcome domains of importance with a prosthesis in LMICs. Further research is required to identify and understand important domains in different social and culture settings, as well as exploring how they vary between countries. This is of particular importance as it is estimated that 80% of the world's population living with a disability live in LMICs [98], and the Global Burden of Disease study 2019 indicates an increasing international amputation prevalence of 176 million [99]. Previous outcome measure consensus work in prosthetics, undertaken by ISPO, also highlighted that many of the measurement tools for use following amputation have been developed in high income countries and call for development of measures suited to LMICs [8]. However, without first understanding which domains are most important to measure in these settings, outcome measure developers may struggle to capture what is meaningful to patients.

## Limitations

When considering the findings of this review it is important to understand that the domains identified in the analysis have been generated from studies with a range of quality scores. Due to previously described issues with critical appraisal as part of systematic reviews of qualitative literature [25,29,30,100], no studies were excluded but were scored and ranked. Although this a common approach used in qualitative syntheses, it is not how the CASP tool was intended for use and should be viewed with caution.

A further limitation of the review is the potential for confirmation bias within the analysis as the authors pre-existing conceptual model was used to inform the 'a priori' framework. Steps were taken to minimise the risk of shoehorning data into the framework by carrying out open line by line coding as the first step in the analysis process and undertaking a separate thematic analysis on data which did not fit easily into the framework, which was then used to further develop the model. A reflexive journal was also kept throughout by the lead author to critically consider methodological and analytical decisions.

The inclusion of only peer reviewed publications written in English led to a single relevant paper being excluded which may have contributed to the findings. The decision was taken not to use translation software as this may have altered the meaning of quotations.

The review also took a broad approach to the search strategy, identifying studies which explored the experiences of lower limb prosthetic users, as few studies were available describing outcome domains of importance. This resulted in inclusion of studies exploring a wide variety of phenomena. It is possible that domains of importance may have been overlooked as none of the studies set out to explore meaningful recovery following prosthetic rehabilitation. However, this wide focus ensured comprehensive inclusion of the available evidence using the research question as a compass rather than an anchor [101]. This facilitated an exploratory approach to understanding outcomes of importance, which is more aligned to primary qualitative methods. Nonetheless It should be considered that researcher judgement was required to identify data presented in the included studies which were relevant to the research question, and required researchers to view the data through a different lens than was originally intended, potentially reinterpreting its meaning.

## Conclusion

This synthesis of qualitative findings from 40 studies representing the views of nearly 600 people provides a rigorous foundation for understanding outcome domains of importance following lower limb prosthetic rehabilitation. By focusing on the patients perspective, the ECLIPSE model attempts to portray a meaningful recovery in the lives of those with limb loss, particularly in high income settings.

The ECLIPSE model is an accessible patient-centred view of recovery and could be used by clinicians to shape and direct the focus of rehabilitation programmes and inform goal setting, as well as direct the evaluation of their impact through the selection of outcome measures. The apparent interconnected nature of outcome domains of importance also highlights the need for a holistic approach to outcome measurement, capturing success in all aspects of the patient's life.

The domains which comprise the ECLIPSE model could also inform the selection of outcomes within research. They could underpin a future core outcome set (COS) or represent a standalone patient COS, which may be more appropriate for rehabilitation settings where the aim is to enable return to previous lives. Future work is needed to understand how well current outcome measures capture the domains described in the model and whether new measures need to be developed.

## Supporting information

**S1 Checklist. PRISMA 2020 checklist.**
(DOCX)

**S1 Appendix. Pre-existing conceptual model.** The Authors previously developed conceptual model of outcome domains of importance following lower limb prosthetic rehabilitation.
(DOCX)

**S1 File.**
(XLSX)

**S2 File.**
(XLSX)

## Author Contributions

**Conceptualization:** Chantel Ostler, Alex Dickinson, Maggie Donovan-Hall.

**Data curation:** Chantel Ostler.

**Formal analysis:** Chantel Ostler, Alex Dickinson, Maggie Donovan-Hall.

**Funding acquisition:** Chantel Ostler, Alex Dickinson, Maggie Donovan-Hall.

**Investigation:** Chantel Ostler.

**Methodology:** Chantel Ostler, Alex Dickinson, Maggie Donovan-Hall.

**Project administration:** Chantel Ostler.

**Resources:** Chantel Ostler.

**Software:** Chantel Ostler.

**Supervision:** Alex Dickinson, Cheryl Metcalf, Maggie Donovan-Hall.

**Validation:** Chantel Ostler.

**Visualization:** Chantel Ostler.

**Writing – original draft:** Chantel Ostler, Maggie Donovan-Hall.

**Writing – review & editing:** Alex Dickinson, Cheryl Metcalf.

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
