## [Decision Letter · Decision Letter 0]

23 Apr 2024

PONE-D-23-42711Development of the ECLIPSE model of meaningful outcome domains following lower limb amputation and prosthetic rehabilitation, through systematic review and best fit framework synthesisPLOS ONE

Dear Dr. ostler`,

Thank you for submitting your manuscript to PLOS ONE. After careful consideration, we feel that it has merit but does not fully meet PLOS ONE’s publication criteria as it currently stands. Therefore, we invite you to submit a revised version of the manuscript that addresses the points raised during the review process.

The reviewers and the academic editor agree that this manuscripts has merits. However, some minor changes are still required, mainly to improve clarity regarding the methods and the delination between previous and new framework. Please address all reviewers' comments. At this stage, please also identify the reviewers XX and XX in the manuscript, and provide a clear statement on contributions of all authors.

We look forward to receiving your revised manuscript.

Kind regards,

Heike Vallery

Academic Editor

PLOS ONE

 [AD -  funding from EPSRC/NIHR (grant ref EP/R014213/1). Funders played no role in the study

CO - University of Southampton’s Institute for Life Sciences. Funders played no role in the study].  

3. We note that Figure 1 in your submission contain copyrighted images. All PLOS content is published under the Creative Commons Attribution License (CC BY 4.0), which means that the manuscript, images, and Supporting Information files will be freely available online, and any third party is permitted to access, download, copy, distribute, and use these materials in any way, even commercially, with proper attribution. For more information, see our copyright guidelines: http://journals.plos.org/plosone/s/licenses-and-copyright.

4. We note that Figure 2 in your submission contain [map/satellite] images which may be copyrighted. All PLOS content is published under the Creative Commons Attribution License (CC BY 4.0), which means that the manuscript, images, and Supporting Information files will be freely available online, and any third party is permitted to access, download, copy, distribute, and use these materials in any way, even commercially, with proper attribution. For these reasons, we cannot publish previously copyrighted maps or satellite images created using proprietary data, such as Google software (Google Maps, Street View, and Earth). For more information, see our copyright guidelines: http://journals.plos.org/plosone/s/licenses-and-copyright.

5. Please upload a copy of Figure 4, to which you refer in your text on page 34. If the figure is no longer to be included as part of the submission please remove all reference to it within the text.

Reviewers' comments:

Reviewer's Responses to Questions

**Comments to the Author**

1. Is the manuscript technically sound, and do the data support the conclusions?

Reviewer #1: Yes

Reviewer #2: Yes

2. Has the statistical analysis been performed appropriately and rigorously? 

Reviewer #1: Yes

Reviewer #2: N/A

3. Have the authors made all data underlying the findings in their manuscript fully available?

Reviewer #1: Yes

Reviewer #2: Yes

4. Is the manuscript presented in an intelligible fashion and written in standard English?

Reviewer #1: Yes

Reviewer #2: Yes

5. Review Comments to the Author

Reviewer #1: Dear authors,

I read your manuscript with great interest and have the following questions that need to be addressed:

- Line 145: The authors searched for 2011 and early 2023. Can this data be specified so that when others reproduce your search, they will get the same results?

- Lines 154-155: Initially, the authors state that two researchers conducted screening. Why did only one author conduct the full-text screening?

- Table 4: How the pre-existing model was developed is unclear.

• What sample was used to develop this framework?

• How was the model developed?

- Line 228: the authors use the PRISMA diagram. Please state that you have used the PRISMA guidelines in the method section.

- Results: Can the different studies be compared to each other? Is the confirmability of the various studies sufficiently large?

- Discussion and conclusion:

• The authors present the ECLIPSE model based on a literature review. Can the results be generalised, given the variety of included studies?

• The authors state that most of the factors included in the framework intertwine. Does the framework then cover the patient perspectives sufficiently? Should the framework be adapted also to include the interaction between the different domains? How does this influence the credibility of the results?

• Overall, the authors should be cautious concerning statements about applying the framework.

- The methods, results and discussion sections have been evaluated by applying Truthworthiness from O’Reilly, M., & Kiyimba, N. (2015). Advanced qualitative research. (Vols. 1-0). SAGE Publications Ltd, https://doi.org/10.4135/9781529622782

Reviewer #2: This article develops a qualitative framework to define and structure relevant outcomes for patients after lower limb amputation. This article builds on the authors' preliminary work on a framework that was initially developed with few (37) and only UK patients. In this paper, an extended framework is derived from a systematic literature review and a synthesis of the qualitative findings from the relevant publications. This new model is based on 40 studies involving 539 patients. This makes the database much broader and more representative (subjects from 15 countries), although the authors point out the (too) few studies from low-income countries, which could introduce a systematic bias. On the basis of these data, the authors have extended their original model.

The work is highly relevant, as the definition of outcomes and then their measurement is of great importance for health care. The inclusion of the perspective of those affected - in this case people with amputations - is particularly important, as only they can assess the subsequent relevance for their everyday life and participation. This perspective also determines the qualitative approach chosen by the authors.

The initial problem definition and the systematics of the study are well presented by the authors. The methodology is appropriate to the problem and the main methodological steps and interim results are well described. The assessment of the quality of the included studies is also interesting, although the exact impact of the assessment on the further analysis/synthesis is not entirely clear.

The actual synthesis consists of two parts: i) the collection of evidence for the existing framework and its domains and ii) the addition of additional sub-themes to the framework. Regarding i), the authors found sufficient evidence to support the framework already developed and therefore further development seems reasonable. Regarding 2), while only minor adjustments were made to some domains (2 and 4), domain 1 was expanded to include the relevant roles of those affected and domain 5 was expanded to include additional aspects of subjective coping and acceptance of 'my new normal'. Domain 3 was split into two old and one new sub-theme. However, sub-theme 3.3 (My prosthesis enables me to participate) does not seem to be clearly distinct from domains 1 and 2. The description and quotes seem to focus more on subjective trust in the prosthesis and the steps involved in building trust.

Overall, the existing framework becomes more comprehensive and differentiated due to the broader database. However, when describing the new aspects, the previous framework should be more clearly delineated, otherwise the additions are not comprehensible and their relevance is not clear.

6. PLOS authors have the option to publish the peer review history of their article (what does this mean?). If published, this will include your full peer review and any attached files.

Reviewer #1: No

Reviewer #2: No

---

## [Author Response · Author response to Decision Letter 0]

10 Jun 2024

All responses to reviewers and editor comments have been documented in detail in a table uploaded in the Response to Reviewers file

---

## [Editor Report · Decision Letter 1]

8 Jul 2024

Development of the ECLIPSE model of meaningful outcome domains following lower limb amputation and prosthetic rehabilitation, through systematic review and best fit framework synthesis

PONE-D-23-42711R1

Dear Dr. ostler`,

We’re pleased to inform you that your manuscript has been judged scientifically suitable for publication and will be formally accepted for publication once it meets all outstanding technical requirements.

Kind regards,

Heike Vallery

Academic Editor

PLOS ONE

Additional Editor Comments (optional):

The formatting of table 1 is deficient.
---

## [Editor Report · Acceptance letter]

13 Jul 2024

PONE-D-23-42711R1 

PLOS ONE

Dear Dr. Ostler, 

I'm pleased to inform you that your manuscript has been deemed suitable for publication in PLOS ONE. Congratulations! Your manuscript is now being handed over to our production team.

Kind regards, 

on behalf of

Dr. Heike Vallery 

Academic Editor

PLOS ONE